# The Interface Microstructures and Mechanical Properties of Laser Additive Repaired Inconel 625 Alloy

**DOI:** 10.3390/ma13194416

**Published:** 2020-10-03

**Authors:** Yiyun Wei, Guomin Le, Qingdong Xu, Lei Yang, Ruiwen Li, Wenyuan Wang

**Affiliations:** Institute of Materials, China Academy of Engineering Physics, Mianyang 621908, China; weiyiyun2020@163.com (Y.W.); leguomin@126.com (G.L.); yang3410@163.com (L.Y.); ruiwenli@163.com (R.L.); wywang@alum.imr.ac.cn (W.W.)

**Keywords:** laser additive repairing, Inconel 625 alloy, interface microstructures, micro-mechanics

## Abstract

The microstructure and micro-mechanics around the repaired interface, and the tensile properties of laser additive repaired (LARed) Inconel 625 alloy were investigated. The results showed that the microstructure around the repaired interface was divided into three zones: the substrate zone (SZ), the heat-affected zone (HAZ), and the repaired zone (RZ). The microstructure of the SZ had a typical equiaxed crystal structure, displaying simultaneously precipitated block-shaped MC-type carbides (NbC, TiC), with bimodal sizes of approximately 10 μm and 0.5 μm and an irregularly shaped flocculent Laves phase. Recrystallization occurred in the HAZ, and led to significant grain growth; a portion of the second phase dissolved in the original grain boundaries. In the RZ, there was a columnar crystal structure, and the size increased with increasing deposition thickness. Moreover, the microstructure between the layer interface and layer interior was quite different, presenting an overlapping transition zone (OTZ), in which the dendritic structure coarsened and more Laves phase were precipitated, compared to in the layer interior. The hardness and tensile properties of the LARed samples were equivalent to those of the wrought substrate, which indicates that laser additive repairing (LAR) is a reliable repair solution for damaged and mis-machined components comprising Inconel 625 alloy.

## 1. Introduction

Inconel 625 is a nickel–chromium solid-solution strengthened alloy, which is largely strengthened by Nb and Mo [1,2]. It is widely used to fabricate high-temperature components in aeronautical, aerospace, marine, chemical, and nuclear industries, due to its extraordinary combination of high yield and tensile strength, excellent creep strength, and outstanding corrosion resistance at elevated temperature [3,4,5]. However, these components are easily damaged owing to their aggressive service conditions, and can also be mis-machined during processing [6]. If these damaged and mis-machined components can be repaired rapidly, considerable savings in materials, processing time, and costs can be achieved.

Laser additive manufacturing (LAM) [7] has attracted much attention in the past ten years. By combining laser cladding with rapid prototyping techniques, LAM is an advanced fabrication technique for producing fully dense near-net-shape metallic components on metallic substrates [8]. Based on LAM, laser additive repairing (LAR) technology has been developed [9,10,11]. By using the damaged or mis-machined components as substrates, the geometry and mechanical properties of the damaged or mis-machined components can be repaired using LAR without deteriorating the performance of the body parts. Compared with traditional repair technologies [12,13,14], such as electrobrush plating, thermal spraying, and argon arc welding, LAR has the advantages of a high degree of automation, a small heat affected zone, metallurgical bonding between the repaired zones and the body part, and a reasonable repair cost [15]. A number of metallic components made of different type of materials, including stainless steel, cobalt-based alloys, titanium alloys, and nickel-based alloys, have been repaired using LAR technology [16,17,18].

In recent years, the processing parameters, microstructures, and mechanical properties of Inconel 625 fabricated using LAM technology have attracted much attention [19,20,21,22]. Dinda et al. [23] studied the microstructural evolution and structural thermal stability of a LAM Inconel 625 alloy. The results indicated that the as-deposited microstructure mostly consisted of columnar dendrites that were stable up to 1000 °C. Rombouts et al. [24] found that Inconel 625 deposited using directed energy deposition (DED) also showed a microstructure of dendrites parallel to the build direction. Rivera et al. [25] observed fine grain structures at the layer interfaces of Inconel 625 produced using solid-state additive manufacturing. Compared to samples fabricated using LAM, not only the microstructures of the repaired part, but also the microstructures of the interface between the substrate and the repaired part, are essential to the mechanical properties of the component repaired using LAR. The research work of LAR for nickel-based alloys mainly focuses on the repair defects, processing parameters, microstructures, and mechanical properties of repaired samples. Onuike et al. [10] used DED technology to repair the internal cracks in Inconel 718 alloy. Sui et al. [9] studied the tensile deformation behavior of LARed Inconel 718 alloy with a non-uniform microstructure. To the best of our knowledge, no studies have investigated the relationship between the microstructure and micro-mechanical properties of LARed Inconel 625 alloy around the repaired interface.

In order to explore the potential of LAR as a reliable repair solution for damaged and mis-machined Inconel 625 components, Inconel 625 alloy substrates, with premade trapezoidal groove shaped defects, were repaired using LAR in this study. The microstructures around the repaired interface and the corresponding mechanical properties were investigated.

## 2. Materials and Methods

### 2.1. Materials

The raw materials were prealloyed Inconel 625 powders prepared using a plasma rotating electrode process. These powders had spherical shapes with sizes of 75~125 μm, as shown in Figure 1a. The microstructure of the powders showed a fine dendritic morphology on the surface, which was caused by the rapid solidification during atomization (Figure 1b). Before the LAR experiment, the powders were dried in a vacuum oven at 120 ± 10 °C for 2 h. The LAR process was performed on heat treated wrought Inconel 625 substrate, for which the chemical composition is listed in Table 1. The chemical composition of Inconel 625 powder is also listed in Table 1.

### 2.2. Experimental Procedures

The LAR experiments were performed on a LAM system (LAM, LSF-12000, Nanjing, China), which consists of a fiber laser with a wavelength of 1070 nm, a five-axis numerical control working table, a powder feeder system with a coaxial nozzle, and an inert atmosphere glove box (oxygen content ≤10 ppm). Argon gas was used to protect the molten pool from oxidation and deliver the alloy powders.

The LAR process was performed on the substrate with a pre-machined groove defect, as shown in Figure 2a. Then the defect was repaired layer by layer using the LAM system, as shown in Figure 2b. Figure 3 shows a schematic of the scanning strategy of the LAR process. The scanning directions in the same layer followed a zigzag pattern, and were rotated by 90 degree in the following layer. The process parameters are listed in Table 2.

A tensile testing bar was cut from the repaired sample by spark erosion. The volume fraction of the repair zone in the cross-section of the tensile testing bar was 50%, as shown in Figure 2c,d.

Microstructures and textures around the repaired interface were examined by optical metallography (OM, Olympus OLS4000, Tokyo, Japan), scanning electron microscopy (SEM, FEI Helios Nanolab 600i, Hillsboro, OR, USA), and electron backscattered diffraction (EBSD, EDAX, Mahwah, NJ, USA). Elemental analysis was carried out using energy-dispersive X-ray spectrometry (EDS, EDAX, Mahwah, NJ, USA). Before the microstructure observation, samples were ground on 500–2000 grit silicon carbide papers and mechanically polished. Polished samples were then electro etched under an 8 V direct current (DC) for 15 s in a reagent of 10% aqueous chromic acid solution to reveal the microstructure. All EBSD samples were polished using an argon beam under a voltage of 5 kV and a current of 110 μA for 20 min before the EBSD test. The EBSD test used a voltage of 20 kV and a step size of 1 μm. X-ray diffraction (XRD, TDF-3000, Dandong, China) scanning was carried out at a constant scanning speed of 10°/min with Cu Kα radiation, and recorded in 2θ ranging from 30° to 100°. To identify tiny phases in the microstructures, a transmission electron microscope (TEM, FEI Titan G2 60-300, Hillsboro, OR, USA) equipped with EDS was applied. The TEM samples, with a 3 mm diameter disc shape, were prepared by mechanical grinding to ~50 μm and then twin-jet polished in a solution of 250 mL HCLO_4_+750 mL methanol at a temperature of −30 °C and a voltage of 25 V. Vickers hardness of the samples was tested under a load of 200 g and a duration time of 15 s along the deposition direction. The elastic modulus and microhardness of different zones were evaluated by nano-indentation (Hysitron TI-950, Minneapolis, MN, USA) under a load of 5 mN. The room temperature tensile properties of the LARed samples and substrates were tested using a floor model universal testing system (CMT5105, Jinan, China), at a displacement rate of 2 mm/min. For each tensile test condition, three samples were used to calculate the average values and standard deviations of strengths, elongations, and area reductions. The ultimate tensile strength, 0.2% offset yield strength and elongation, was obtained from the resulting stress-strain curves, and the area reduction was determined by measuring the cross-sectional area before and after failure. The fracture morphology was observed using SEM.

## 3. Results and Discussion

### 3.1. Microstructure Around the Repaired Interface

#### 3.1.1. Grain Structures

Figure 4a–d show the microstructures in the X–Z section and Y–Z section of the LARed samples around the repaired interface, respectively. The interface between the substrate and the repaired zone presents good metallurgical bonding. No obvious defects, such as cracks and pores can be observed. A transition of microstructure can be observed along the deposition direction, dividing the area into three zones: the substrate zone (SZ), the heat-affected zone (HAZ), and the repaired zone (RZ) (see Figure 4b,d). Inside the RZ, a narrow overlapping transition zone (OTZ) can be observed, between two adjacent deposited layers or tracks (see Figure 4b,d). The width of the OTZ is approximately 0.15 mm. The SZ consists of a mixture of finely and large equiaxed grains. The grains in the HAZ are hard to reveal under the same erosion conditions, suggesting a composition variation in grain boundaries. Detailed grain structure observation was carried out using EBSD, as in the following. RZ presents a columnar dendritic structure that grows epitaxially along the deposition direction. In addition, the dendritic structure in the OTZ between two adjacent deposited tracks is coarser than other areas in the RZ.

Detailed investigations on the microstructures around the repaired interface of the LARed Inconel 625 were carried out using EBSD. Figure 5a–d show the EBSD images in the X–Z section and Y–Z section of the LARed samples around the repaired interface, respectively. The three zones of SZ, HAZ, and RZ can be easily distinguished in the EBSD images, showing grains with different sizes and distributions. The SZ has typical equiaxed grains, with a bimodal grain size distribution. Annealing twins can be observed in the recrystallized grains. Grains in the HAZ grew significantly larger compared to those in the SZ. In the RZ, large sized columnar grains grew epitaxially from the mother grains in the HAZ. The height of the columnar grains can be as large as the layer height. The widths and heights of the columnar grains increase as the deposition height increases. The OTZ cannot be observed in the EBSD maps.

Figure 6a–k show the EBSD results of the SZ and RZ in the X–Y section of the LARed samples, respectively. The SZ has typical equiaxed grains, with a bimodal grain size distribution. Compared to the SZ, the RZ has a larger grain size, and the grain orientation has a stronger texture.

#### 3.1.2. Phases and Precipitations

Figure 7 shows the X-ray diffraction patterns of the RZ and SZ in the repaired sample. It can be demonstrated that the RZ and SZ in the LARed samples both mainly consist of the γ (Ni–Cr) phase without peaks from other precipitates [26], such as γ′ (Ni_3_Al(Ti)), γ″ (Ni_3_Nb), δ (Ni_3_Nb), Laves (Ni, Fe, Cr)_2_(Nb, Ti, Mo), and carbide phases, due to their low volume fractions or absences. Furthermore, the diffraction peak intensity of the (200) crystal plane of the γ phase in the RZ is higher than that of the (111) crystal plane, which also indicates that the microstructure in the RZ has a strong texture.

Detailed SEM investigations were applied to the areas around the repaired interface. Figure 8 shows SEM images of the SZ. A large number of precipitates can be seen in the grain boundaries and grain interior (Figure 8a,b). The precipitates have two typical average sizes of 10 μm and 0.5 μm. The large precipitates mainly have irregular shapes. The small precipitates have both flocculent shapes and polygonal shapes (see Figure 8c). The chemical compositions of the matrix (as indicated by point 1 in Figure 8a) and large sized precipitates (as indicated by points 2–4 in Figure 8a) were analyzed using EDS equipped in the SEM, and the results are shown in Figure 8d. The large precipitates mainly comprised Nb and Ti, while the matrix was rich in Ni and Cr but depleted in Nb and Ti. The results indicate that Nb and Ti in the matrix were segregated to form large precipitates. These large precipitates were assumed to be MC-type carbides (M is Nb or Ti), while the matrix was assumed to be a γ (Ni-Cr) solid solution. TEM observations and the equipped EDS X-ray microanalysis were carried out to clarify the small precipitates (shown in Figure 8e,f). The EDS analysis results revealed that the polygonal shaped precipitate was NbC, and the flocculent phase was a Laves phase, which is consistent with the observations made by Xu [27].

Figure 9a presents the microstructure of repaired samples around the repaired interface. The microstructure is quite different in the SZ and the RZ along the deposition direction. Similarly to the SZ, a large amount of white precipitates are also seen in the HAZ and RZ. From the previous analysis, we can see that the precipitates of the SZ mainly consist of large (approximately 10 μm) irregular shaped MC-type carbides (M is Nb and Ti), small (approximately 0.5 μm) polygonal shaped MC-type carbides, and flocculent shaped Laves phase. In the HAZ, there are still many large irregular shaped MC-type carbides, but some small sized precipitates dissolved in the original grain boundaries, as shown in Figure 9b.

For the RZ, the microstructure indicates epitaxial growth of the columnar dendritic structure, and a large amount of white precipitates occur along the dendritic boundaries, as shown in Figure 9a. In addition, compared to the microstructure in the layer interior, the amount of precipitates is greater, and the size is largest in the OTZ around the repaired interface, as shown in Figure 9c,d. The precipitates mainly have flocculent shapes. Figure 9f shows the EDS analysis results of the flocculent shaped phase and the matrix phase. The results of the EDS analysis reveal that the main hardening elements Nb and Mo are rich in the flocculent shaped phase, which is the characteristic Laves phase of Inconel 625 alloys [28,29]. This indicates a somewhat heavy segregation in the OTZ. Figure 10 further shows the high resolution transmission electron microscopy (HRTEM) image and the EDS results of nanosized precipitates in the RZ. The EDS analysis indicates that it is NbC. Therefore, the precipitates of the RZ mainly consist of the flocculent shaped Laves phase and a small amount of nanosized MC-type carbides.

#### 3.1.3. Microstructure in the Overlapping Transition Zone

Figure 11 shows the microstructure of the OTZ between two adjacent deposited tracks in the RZ. Similarly to the OTZ around the repaired interface, the microstructure of the OTZ in the RZ also presents a coarse columnar dendritic structure in Figure 11a. The width of the OTZ is also approximately 0.15 mm. Particularly, a large amount of white precipitates occur along the dendritic boundaries in the OTZ, as shown in the Figure 11b. The EDS results of the precipitates demonstrate that they are Laves phase, as shown in Figure 11c,d.

The microstructural characteristics between the OTZ and layer interior are quite different, which can be explained by the classical theoretical model of dendrite growth during the solidification process. Together, the temperature gradient, *G*, and the growth rate, *R*, dominate the solidification microstructure. The Kurz, Giovanola, Trivedi (KGT)model has often been utilized to investigate how the *G* and *R* affect the primary dendritic spacing of a deposited metal [30]. According to the KGT model, the variation of dendritic tip radius *r* with the growth rate *R* of a Ni-based alloy can be described as follows [31]:(1)$$r=\pi \sqrt{\frac{4D_{L}\Gamma }{k\Delta TR}},$$
where *Γ* is the Gibbs–Thomson coefficient, *D_L_* is the liquid inter-diffusion coefficient, *k* is the equilibrium diffusion coefficient, Δ*T* is the non-equilibrium solidification range. Furthermore, according to the Kurz and Fisher analysis [32], the primary dendritic spacing *λ* can be expressed by a function of the dendritic tip radius *r* as [33]:(2)$$\lambda =\sqrt{\frac{3\Delta Tr}{G}},$$

Therefore, the primary dendritic spacing *λ* is identified as a function of temperature gradient *G* and the growth rate *R* as follows:(3)$$\lambda ={\left(\frac{3\pi }{G}\right)}^{1/2}{\left(\frac{4D_{L}\Delta T\Gamma }{kR}\right)}^{1/4}\propto G^{-1/2}R^{-1/4},$$

The product *G*·*R* (cooling speed) governs the size of the solidification structure [34]. Thus, the difference in the temperature gradient *G* and the growth rate *R* during the LAR process should be the primary reason for the different morphologies in the OTZ. During the LAR process, the material is deposited track-by-track and layer-by-layer. When a new track is deposited, the top portion of the material in the previous track is remelted, and then forms a molten pool, as shown in Figure 12. The temperature of the previous track is higher than that of the previous layer, therefore, the temperature gradient at the overlapping line, *G*_1_, is less than at the other fusion line, *G*_2_. Consequently, the *G*·*R* (cooling rate) is lower at the overlapping line and higher at the other fusion line. This leads to a significant increase in the primary dendritic spacing *λ* along the overlapping line of the molten pool, which forms an OTZ between tracks.

The size of the Laves phases is attributed to the concentration of the alloying elements. During the LAR process, solidification in the Inconel 625 alloy starts with the primary liquid→γ reaction, causing the accumulation of Nb, Mo, C, and Ti in the interdendritic and grain boundary liquids. Thus, the Laves phase and MC-type carbides can precipitate in these regions. Then, the subsequent liquid → (γ + NbC) eutectic reaction consumes most of the carbon available, until another eutectic reaction L→ γ + Laves occurs, finishing the solidification process [27]. The Laves phase is an unavoidable terminal solidification phase in Inconel 625 alloys. However, solidification conditions can strongly influence the extent of niobium segregation and the amount of Laves phase [35]. Thus, a slow cooling rate (*G*·*R*) in the OTZ causes a large segregation of the alloying elements, which leads to the formation of a large amount of Laves phase.

### 3.2. Composition Distribution

The EDS line scanning results show that the major alloying elements, such as Ni and Cr, are distributed uniformly from the SZ to the RZ, as shown in Figure 13. The fluctuations of some elements are mainly due to the detections of precipitations. However, the Nb and Mo elements are distributed more uniformly in the RZ compared to in the SZ. This is mainly due to the segregation of Nb and Mo elements in the SZ to form a large amount of MC-type carbide and Laves phases, while the degree of segregation of the alloying elements in the RZ is weak, and the precipitated phase is low. In addition, the degree of segregation of the alloying elements is aggravated in the OTZ, compared to in the layer interior, this is due to Nb and Mo elements segregating and forming a large amount of Laves phase.

### 3.3. Mechanical Properties

#### 3.3.1. Hardness

Figure 14 shows the Vickers hardness along the deposition direction, around the repaired interface. The hardness values have no notable difference between the SZ and RZ. The average hardness is approximately 240 ± 20 HV.

The micro-mechanical properties of the LARed Inconel 625 alloy samples were evaluated using a nanoindentation for the reduced modulus (*E_r_*) and microhardness (*H*) around the repaired interface. The nanoindentation load-depth curves of the matrix phase in different zones of the LARed samples are shown in Figure 15a. The responses of the LARed Inconel 625 alloy are predominantly plastic, although some elastic recovery upon release of the load can also be observed. In both figures, a small plateau can be seen at the maximum load, which is indicative of creep in the Inconel 625 alloy.

The resultant load-depth curves were used to obtain the *H* and *E_r_* by employing the procedure of Oliver and Pharr [36]. The *E_r_* was then used, together with Poisson’s ratio (*ν*), to calculate the elastic modulus (*E*) from the following equation:(4)$$\frac{1}{E_{r}}=\frac{1-\nu^{2}}{E}+\frac{1-\nu_{i}^{2}}{E_{i}},$$
where *E_i_* is the elastic modulus of the indenter (1140 GPa) and *ν_i_* is its Poisson’s ratio (0.07) [37]. Furthermore, the Poisson’s ratio used for the Inconel 625 alloy was set to 0.303 [38]. The obtained indentation modulus (*E_r_*) and microhardness (*H*) data of the LARed Inconel 625 alloy sample in different zones are shown in Figure 15b. The *E_r_* values for different zones reveal imperceptible changes. The microhardness of the matrix phase in the RZ is approximately 5.1 GPa, which is slightly higher than that of the SZ (approximately 4.8 GPa). This is due to the weakening of the segregation of the Nb and Mo alloying elements in the RZ. In addition, the microhardness of the matrix phase in the HAZ generally demonstrates a lower value (approximately 4.6 GPa). This is attributed to the dissolution of the MC-type carbide and Laves phases and the growth of the grains in the HAZ. In particular, the microhardness of the matrix phase in the OTZ is approximately 4.7 GPa, which is slightly lower than that of layer interior in the RZ (approximately 5.1 GPa). The formation of abundant Laves phases in the OTZ is due to the large segregation of alloying elements Nb and Mo. The Laves phase is a brittle harmful phase, and consumes a large amount of Nb and Mo, and decreases the amount of Nb and Mo in the matrix, which leads to the decrease of the microhardness of the matrix phase in the OTZ.

#### 3.3.2. Tensile Tests

The room temperature tensile properties of the LARed sample and the heat treated wrought substrate are listed in Table 3. The tensile strength and ductility of LARed samples are similar to those of the wrought substrate, which indicates that LAR is a reliable approach for damaged and mis-machined components made of the Inconel 625 alloy.

To compare the tensile behaviors of the samples in the various conditions, tensile testing of the LARed and wrought samples was done, and the results are shown in Figure 16. The tensile test curves of the different samples are almost the same, which indicates that they have similar tensile properties. The LARed sample demonstrates a slightly steeper elastic response than the wrought sample and yields at a different stress level. After yielding, the tensile specimens in all cases show almost parallel stress–strain curves in the plastic elongation period before fracture.

Figure 17 shows the fractographs of the wrought sample. The fracture surface of the wrought sample exhibits fine dimples, which indicate a ductile mode of failure associated with good tensile properties. In addition, a large number of secondary cracks can be found. The fractograph of the wrought sample shows a distinct morphology, with MC-type carbides inside the larger dimples. At the same time, the second phase MC-type carbides are torn, as shown in Figure 17b. It is evident that the MC-type carbides are primarily responsible for making the fracture process easier by providing favorable sites for excessive microvoid initiation, and the growth of macroscopic cracks.

Figure 18 shows the fractographs of the LARed sample. A mixed fracture occurred in the samples, which included the SZ and RZ, and the fractographs are quite different, as shown in Figure 18a. Figure 18b shows the high-magnification SEM image of the SZ, which is similar to the wrought sample, where the fracture of the SZ contains a dimpled surface indicative of ductile failure, and there are particles inside the large dimples. The dimples and tear edges in the RZ fracture have a clear orientation, as shown in Figure 18c,d. Since the direction of the tensile stress at this time is almost perpendicular to the direction of the dendrite growth in the sample, the aligned dimples and tear edges also indicate that the microstructure of the LARed samples has a distinct orientation in the deposition direction. When the sample is stretched perpendicularly to the deposition direction, the dendritic structure is cut transversely, leaving a dimpled structure, with the dendrite center and dendrite dry areas as the torn edge. As the dendrite growth and dendrite column arrangement is regular, a rupture is left after the regular arrangement of the dimples.

## 4. Conclusions

In this paper, Inconel 625 alloy substrates with premade trapezoidal groove shaped defects were repaired using LAR. The microstructures around the repaired interface and the corresponding mechanical properties were investigated. The conclusions are outlined below:The microstructure around the repaired interface of a LARed Inconel 625 alloy can be divided into three zones: the SZ, HAZ, and RZ. The SZ has a typical equiaxed crystal structure with a bimodal grain size distribution, while in the HAZ, recrystallization occurs and leads to significant grain growth. In the RZ, there are very large columnar grains, and the size of the columns increases with an increase in the number of deposited layers.The precipitates in the SZ mainly consist of large (approximately 10 μm) and small (approximately 0.5 μm) block-shaped MC-type carbides (M is Nb and Ti) and irregularly shaped flocculent Laves phase. In the HAZ, there are still many large block-shaped MC-type carbides, but some precipitates dissolve in the original grain boundaries. For the RZ, some Laves and few MC-type carbides are precipitated along the epitaxial growth dendritic boundaries.The microstructure between two adjacent deposited tracks presented an overlapping transition zone (OTZ), which the dendritic structure coarsened, and more Laves phase precipitated compared to in the layer interior. The width of the OTZ was approximately 0.15 mm. The lower G·R in the OTZ led to an increase in dendrite arm spacing and the formation of the Laves phase.The Vickers hardness and indentation modulus from the SZ to the RZ along the deposition direction were not notably different in the LARed samples, and were approximately 240 ± 20 HV. The microhardness of the matrix phase in the RZ (approximately 5.1 GPa) was slightly higher than that of the SZ (approximately 4.8 GPa), while the microhardness generally achieved a lower value (approximately 4.6 GPa) in the HAZ. In particular, the microhardness of the matrix phase in the OTZ was approximately 4.7 GPa, which was slightly lower than that of layer interior in the RZ (approximately 5.1 GPa). The fluctuation of the microhardness of the matrix phase around the repaired interface was mainly caused by the segregation of the Nb and Mo alloying elements.The yield strength and elastic modulus of the LARed samples were higher than those of the wrought sample. The tensile strength and ductility of the LARed samples were similar to those of the wrought sample. Both the SZ and RZ presented a dimple fracture surface, and a large number of secondary cracks could be found in the SZ. The dimples and tear edges had a clear orientation in the RZ. The comprehensive tensile properties of the LARed Inconel 625 alloy are equivalent to those of the wrought alloy.

## Figures and Tables

**Figure 1 materials-13-04416-f001:**
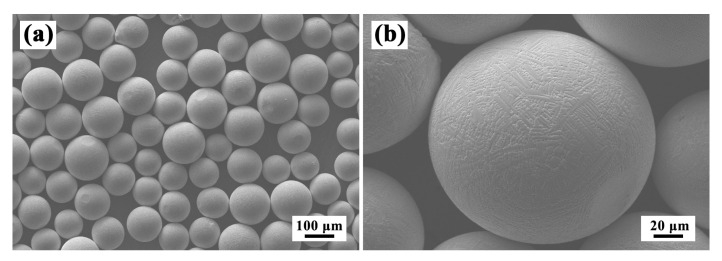
Morphologies of Inconel 625 powders at (**a**) low and (**b**) high magnifications.

**Figure 2 materials-13-04416-f002:**
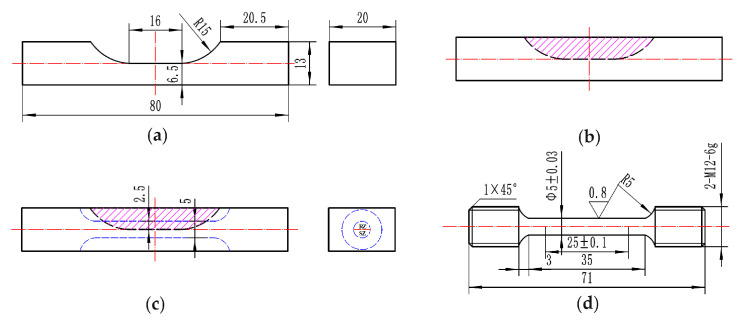
Sketch of (**a**) substrate with pre-machined groove defect, (**b**) LARed sample, (**c**) the position of the tensile testing bar in the LARed sample, and (**d**) shape and size of the tensile testing bar.

**Figure 3 materials-13-04416-f003:**
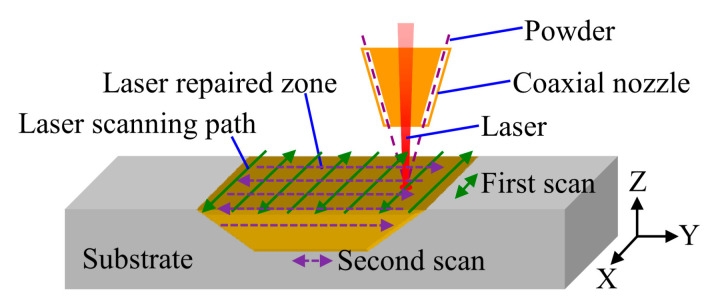
Schematic of the scanning strategy of the LAR process.

**Figure 4 materials-13-04416-f004:**
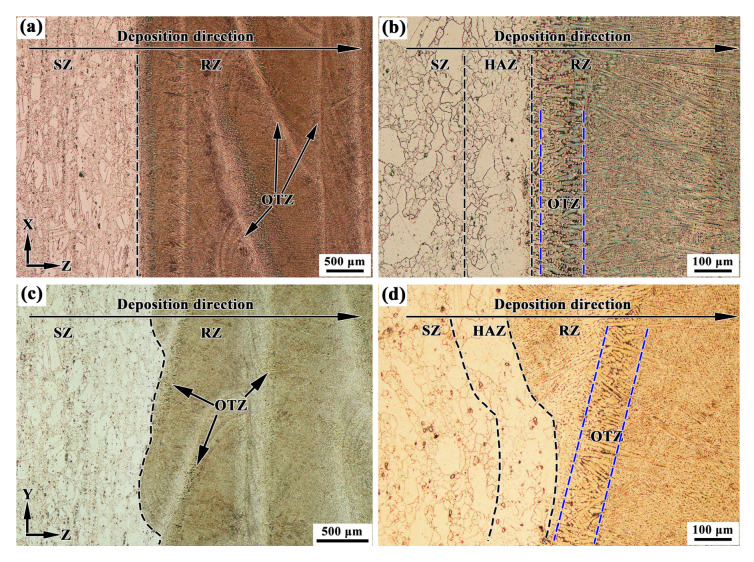
OM graphs in the (**a**,**b**) X–Z section and (**c**,**d**) Y–Z section of LARed Inconel 625 samples around the repaired interface.

**Figure 5 materials-13-04416-f005:**
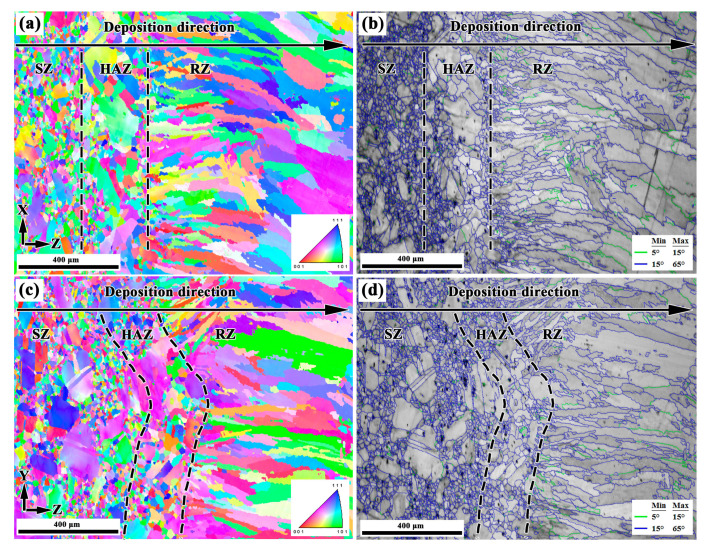
EBSD results of the LARed Inconel 625 samples around the repaired interface in the (**a**,**b**) X–Z section, and (**c**,**d**) Y–Z section: (**a**,**c**) inverse pole figure (IPF) orientation maps, and (**b**,**d**) band contrast maps.

**Figure 6 materials-13-04416-f006:**
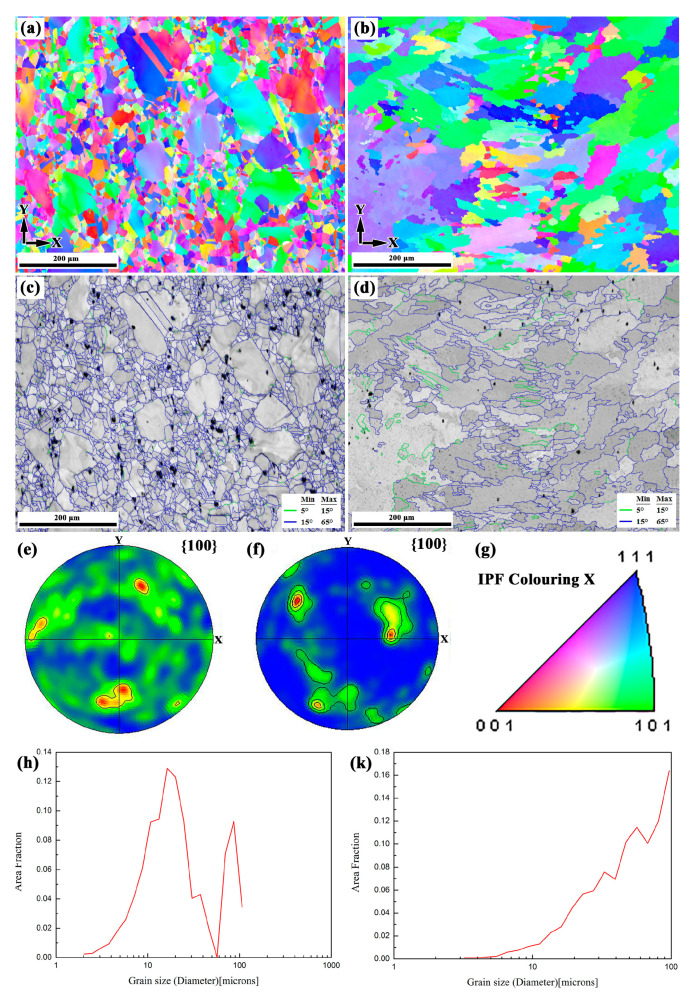
EBSD results of LARed Inconel 625 samples of the substrate zone (SZ) (**a**,**c**,**e**,**h**) and the repaired zone (RZ) (**b**,**d**,**f**,**k**) in the X–Y section: (**a**,**b**) IPF orientation maps, (**c**,**d**) band contrast maps, (**e**,**f**) {100} pole figures, (**g**) IPF and (**h**,**k**) grain size distribution.

**Figure 7 materials-13-04416-f007:**
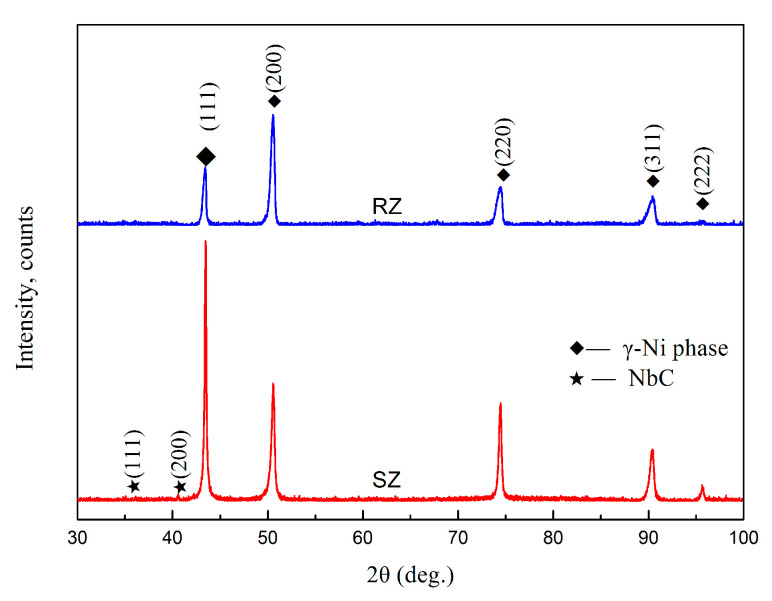
XRD patterns of SZ and RZ in the LARed samples.

**Figure 8 materials-13-04416-f008:**
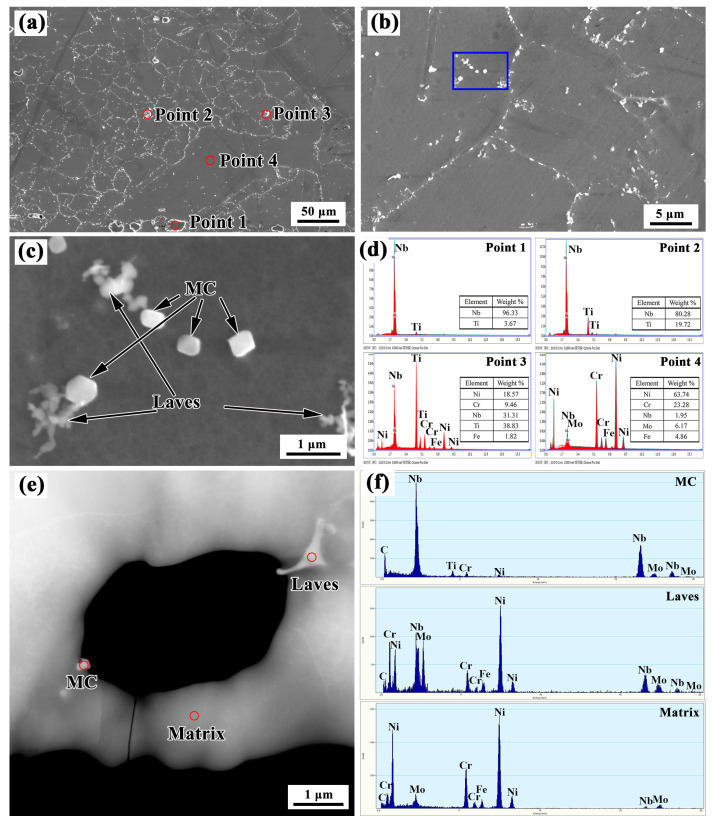
(**a**–**c**) SEM images, (**e**) transmission electron microscope (TEM) images, and (**d**,**f**) energy-dispersive X-ray spectrometry (EDS) results in the SZ.

**Figure 9 materials-13-04416-f009:**
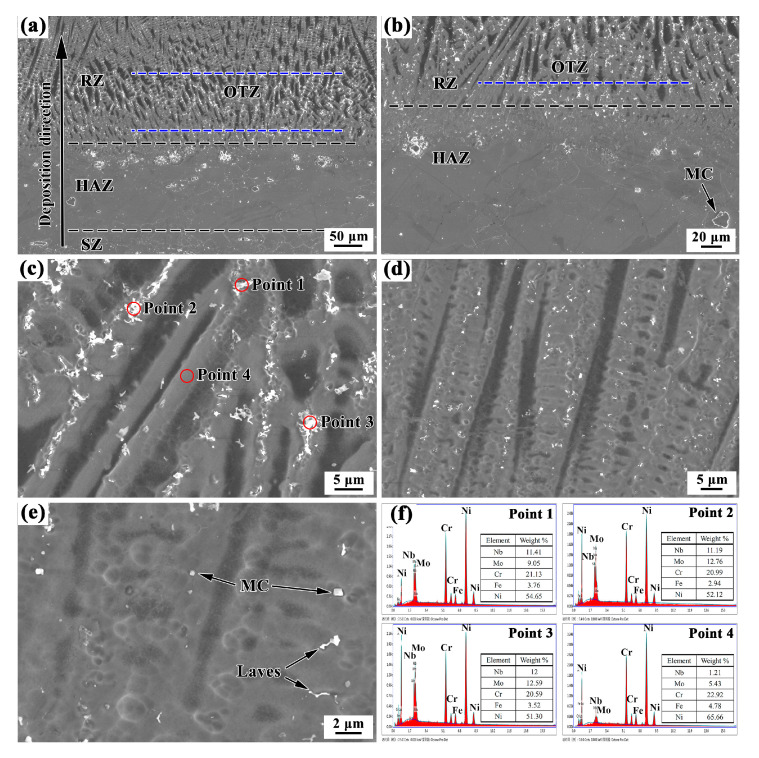
Microstructures of LARed samples in different zones: (**a**) SZ+heat-affected zone (HAZ)+RZ, (**b**) HAZ+RZ, (**c**) overlapping transition zone (OTZ), (**d**,**e**) RZ, and (**f**) EDS taken from laves phase and matrix phase of OTZ around the repaired interface.

**Figure 10 materials-13-04416-f010:**
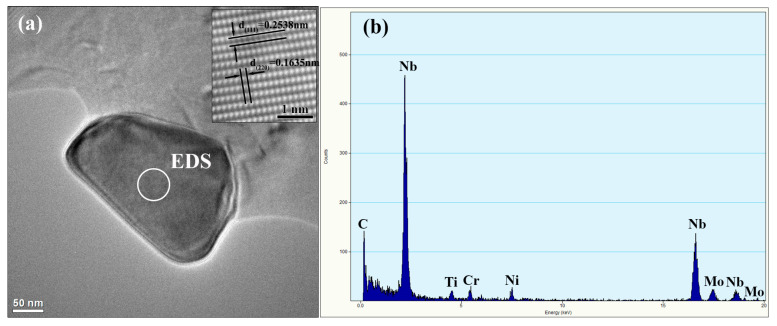
HRTEM images (**a**) of LARed samples in the RZ, and (**b**) the EDS results of a NbC precipitate.

**Figure 11 materials-13-04416-f011:**
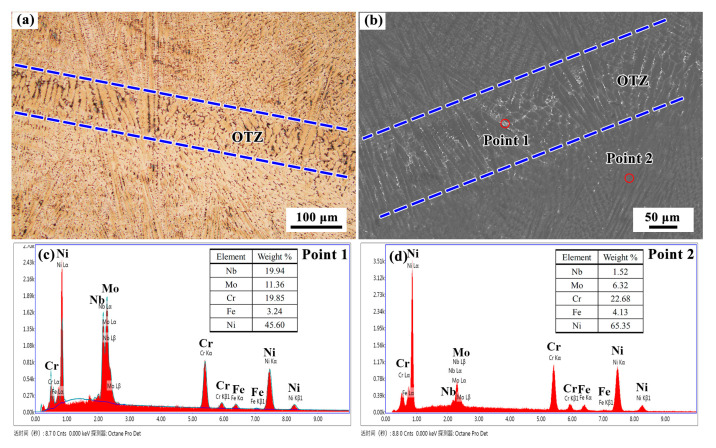
(**a**,**b**) Microstructures of OTZ and the (**c**) EDS results of Laves and (**d**) matrix phases.

**Figure 12 materials-13-04416-f012:**
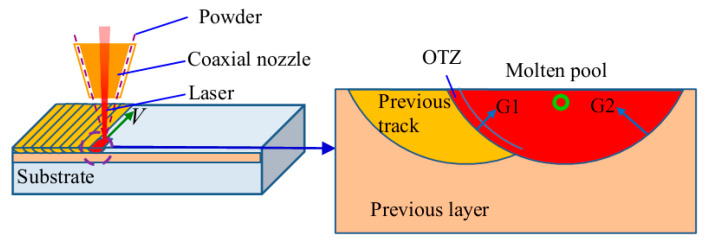
Schematic illustration of the temperature gradient, G, and growth rate, R, distribution in the molten pool during the LAR process.

**Figure 13 materials-13-04416-f013:**
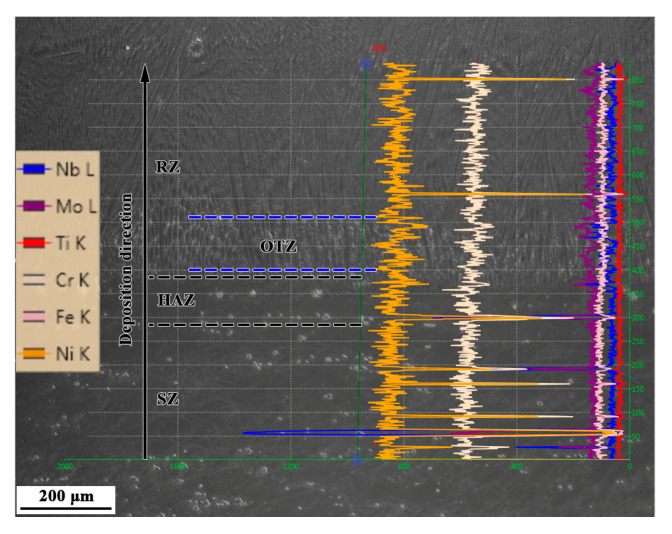
EDS line scanning results of the LARed Inconel 625 alloy sample along the deposition direction.

**Figure 14 materials-13-04416-f014:**
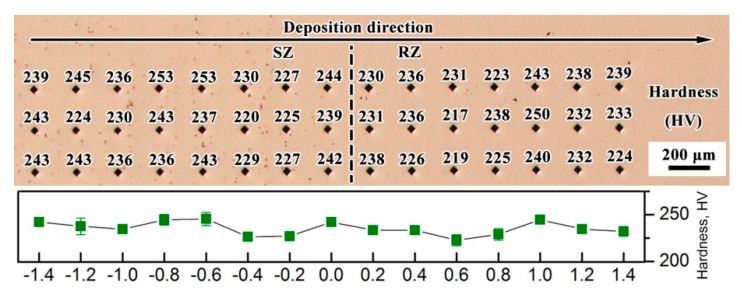
Vickers hardness values of the LARed Inconel 625 alloy sample.

**Figure 15 materials-13-04416-f015:**
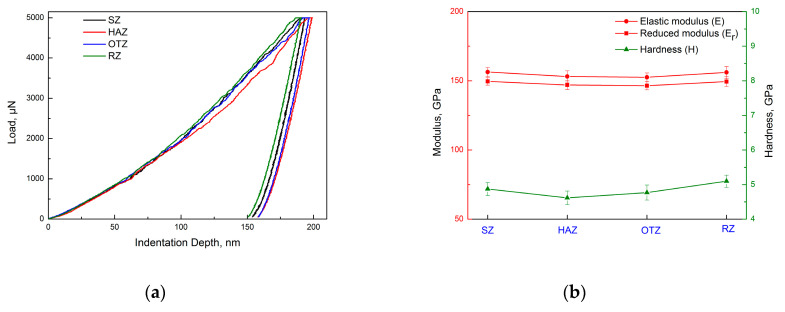
(**a**) Nanoindentation load–depth curves and (**b**) indentation modulus and hardness data of the matrix phase in different zones of the LARed samples.

**Figure 16 materials-13-04416-f016:**
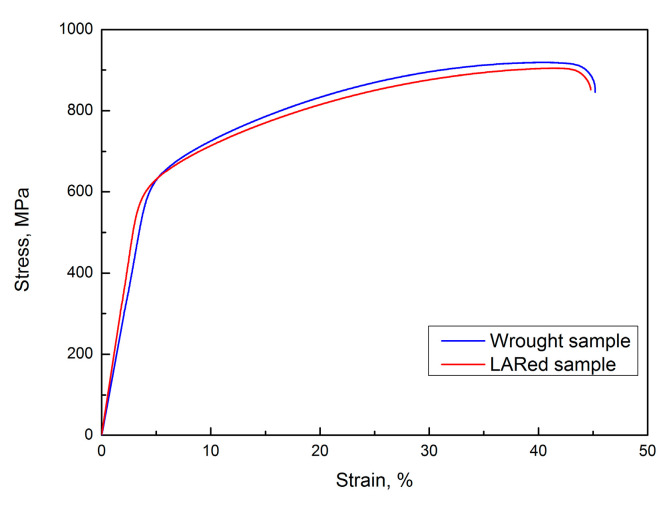
Typical tensile test curves of LARed sample and wrought substrate.

**Figure 17 materials-13-04416-f017:**
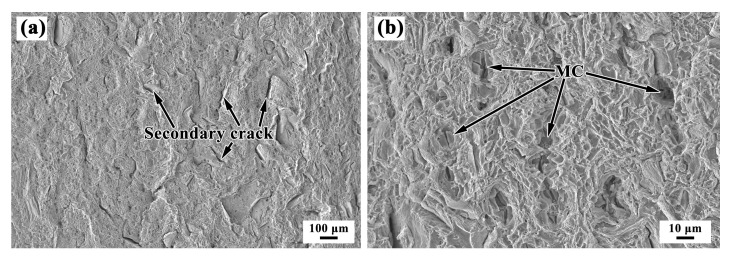
Fractographs of the wrought substrate sample at (**a**) low and (**b**) high magnifications.

**Figure 18 materials-13-04416-f018:**
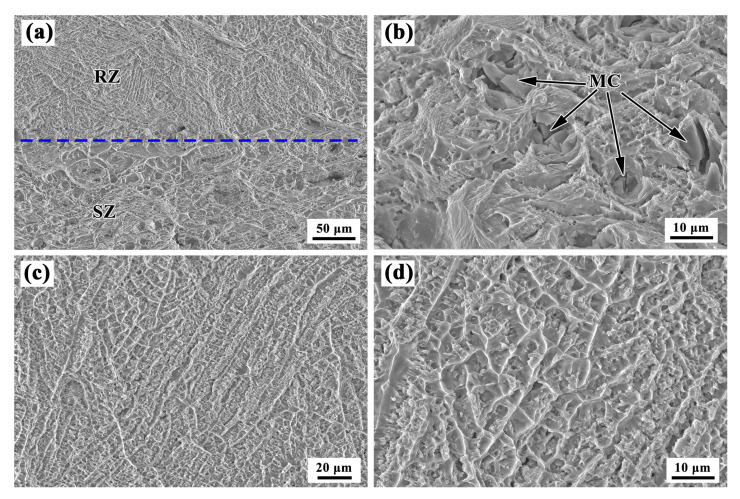
Fractograph of the LARed sample: (**a**) SZ+RZ, (**b**) SZ, and (**c**,**d**) RZ.

**Table 1 materials-13-04416-t001:** The chemical compositions (wt %) of Inconel 625 alloy powder and substrate.

Material	Ni	Cr	Mo	Nb	Fe	Ti	Al	C	Co	Mn	Si	P
Powder	Bal.	21.92	9.09	3.62	3.71	0.20	0.17	0.036	0.009	0.015	0.06	0.003
Substrate	Bal.	21.70	8.90	3.27	4.40	0.18	0.12	0.030	0.010	0.170	0.15	0.008

**Table 2 materials-13-04416-t002:** Process parameters of LARed Inconel 625 alloy.

Laser Power (kW)	Scanning Speed (mm/min)	Laser Spot Diameter (mm)	Increment of Z Axis (mm)	Powder Feeder Rate (g/min)	Overlaps (%)
1.4	400	3	0.5	6	45

**Table 3 materials-13-04416-t003:** Room temperature tensile properties of wrought and LARed Inconel 625 alloy samples.

Number	Ultimate Tensile Strength σ_b_ (Mpa)	Yield Strength σ_0.2_ (Mpa)	Elongation δ (%)	Area Reduction Ψ (%)
LARed sample	908 ± 5	587 ± 3	44.7 ± 0.5	45.0 ± 0.7
Wrought substrate	914 ± 4	572 ± 6	45.3 ± 0.4	43.7 ± 0.3

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
