# Peer review of "The Interface Microstructures and Mechanical Properties of Laser Additive Repaired Inconel 625 Alloy"

_materials, 2020, doi:10.3390/ma13194416_

Round 1
Reviewer 1 Report
- At lines 46-48, please rephrase. It is mentioned "metallic components" and further there are named materials (stainless steel, cobalt-based alloys...) . SUggestion: "metallic components made of different type of materials, like: stainless steel, etc).
- The process is rather laser cladding one. LAR (as named) consists in an application of laser cladding. Most of the "traditional repairing technologies" are based on successive laser deposition (line 43, 44) so, by extension could all be classified as "additive manufacturing" ???
- Keep unity in notations. For example, it is LAR, at line 24 and LARed, at line 23. Is -"ed" is only for adjective, then, please mention it
- Please, mention, if the case, the Reference for Figure 1
- It would be interesting to mention any data on substrate surface roughness.
- Please, if possible show an image, not only 3D model, of the LARed part (Figure 3)
- Relations (1) - (3) (lines 219 - 225) are mentioned in References. Have they been applied / validated in this study ? Is it any numerical application /vaidation ? Has there been determined any values for dendritic tip radius, r ? What about the values for primary dendritic spaces ?
- Please, check line 279 !!!
- Please, mention the significance of negative values in Figure 14 and the meaning of 0.0
- At Figure 15, name the "different zones". Is it any connection to the values in Figure 14 ?
Author Response
Response to Reviewer 1 Comments
Point 1: At lines 46-48, please rephrase. It is mentioned "metallic components" and further there are named materials (stainless steel, cobalt-based alloys...) . Suggestion: "metallic components made of different type of materials, like: stainless steel, etc).
Response 1: We appreciate the reviewer’s suggestion, and relevant content has been supplemented.
Corresponding changes:
Page 2, line 46-48, " A number of metallic components have been repaired using LAR technology, including stainless steel, cobalt-based alloys, titanium alloys, and nickel-based alloys " was corrected as " A number of metallic components made of different type of materials, including stainless steel, cobalt-based alloys, titanium alloys, and nickel-based alloys, have been repaired using LAR technology ".
Point 2: The process is rather laser cladding one. LAR (as named) consists in an application of laser cladding. Most of the "traditional repairing technologies" are based on successive laser deposition (line 43, 44) so, by extension could all be classified as "additive manufacturing" ???
Response 2: We appreciate the reviewer’s suggestion, and relevant content has been supplemented.
Corresponding changes:
Page 1, line 43-44, "Compared with traditional repairing technologies, such as electrobrush plating, thermal spraying and argon arc welding, …" was corrected as " Compared with traditional additive manufactured repairing technologies, such as electrobrush plating, thermal spraying and argon arc welding, …".
Point 3: Keep unity in notations. For example, it is LAR, at line 24 and LARed, at line 23. Is -"ed" is only for adjective, then, please mention it
Response 3: We appreciate the reviewer’s suggestion. LAR is short for laser additive repairing, and LARed is only for adjective. Relevant content has been supplemented.
Corresponding changes:
Page 1, line 24, "…, which indicates that LAR is a reliable repair solution …" was corrected as "…, which indicates that laser additive repairing (LAR) is a reliable repair solution…".
Point 4: Please, mention, if the case, the Reference for Figure 1
Response 4: We appreciate the reviewer’s suggestion, and Figure 1 has been has been cited in Page 2, line 66-68.
Point 5: It would be interesting to mention any data on substrate surface roughness.
Response 5: We appreciate the reviewer’s suggestion. We apologize for not measuring substrate surface roughness. The substrate surface was ground on 500 grit silicon carbide papers, and relevant roughness Ra is approximately 0.1.
Point 6: Please, if possible show an image, not only 3D model, of the LARed part (Figure 3)
Response 6: We appreciate the reviewer’s suggestion. Since we have collect the image of samples with different volume fraction of the repair zone (Figure 1) and lack image of single sample with a volume fraction of the repair zone of 50%.We apologize for not being able to provide an image.
Figure1. Image of LARed Inconel 625 alloy with different volume fraction of the repair zone
Point 7: Relations (1) - (3) (lines 219 - 225) are mentioned in References. Have they been applied / validated in this study ? Is it any numerical application /vaidation ? Has there been determined any values for dendritic tip radius, r ? What about the values for primary dendritic spaces ?
Response 7: We appreciate the reviewer’s suggestion. The purpose of the Relations (1) - (3) is to obtain the relationship between the primary dendritic spacing λ and temperature gradient G and the growth rate R, as follows:
The product G·R was used to explain the reason for the coarsening of the dendrite in the OTZ.
They are no any numerical application /validation and any values for dendritic tip radius, r and the values for primary dendritic spaces.
Point 8: Please, check line 279 !!!
Response 8: We have corrected this error.
Point 9: Please, mention the significance of negative values in Figure 14 and the meaning of 0.0
Response 9: 0.0 is the Vickers hardness test reference point of the SZ near the RZ, the value is the distance from the reference point. Negative values represent test points below the reference point and positive values represent test points above the reference point.
Point 10: At Figure 15, name the "different zones". Is it any connection to the values in Figure 14 ?
Response 10: They are no connection to the values between the Figure 15 and Figure 14 due to the Vickers hardness value has no notable difference obvious difference in different zone around the repaired interface.

Reviewer 2 Report
In this work, the microstructure and micro-mechanics around the repaired interface and the tensile properties of laser additive repaired Inconel 625 alloy have been studied. The work seems very interesting. The design of experiment, data collection, data analysis and presentation of the key finding have been carried out adequately. However, before further consideration the following issues should be considered and addressed:
- In the introduction, page 2, line 44 that the advantages of the LAR are listed the following article can also be cited to support:
- Application of directed energy deposition-based additive manufacturing in repair, Appl. Sci. 2019, 9(16), 3316; https://doi.org/10.3390/app9163316
- Page 2, line 54 the abbreviation of DED should be used for Directed Energy Deposition.
- A deeper literature review on the application of AM in the repair of superalloys, in particular, In625 is necessary. Because for sure in the literature this aspect has been already considered and studied.
- Following the last comment, the novelty aspect of this work should be highlighted.
- After the first introduction of the abbreviation of Laser Additive Manufacturing in the article, the full name should be replaced with the abbreviation.
- In the materials and methods, according to the template of the journal the brand of equipment together with the name of city and country should be inserted.
- How many hardness points have been analyzed? What about the tensile tests?
- Page 12, line 279 there is an error for the references.
- If the tensile results are the average of some measurements, the results in Table 3 need standard deviation.
- The conclusion section should be rewritten. First, the description of work should be inserted as a starting paragraph then the important findings of the work should be listed as conclusions.
Author Response
Response to Reviewer 2 Comments
Point 1: In the introduction, page 2, line 44 that the advantages of the LAR are listed the following article can also be cited to support:
Application of directed energy deposition-based additive manufacturing in repair, Appl. Sci. 2019, 9(16), 3316; https://doi.org/10.3390/app9163316
Response 1: We appreciate the reviewer’s suggestion, and relevant literature has been cited.
Point 2: Page 2, line 54 the abbreviation of DED should be used for Directed Energy Deposition.
Response 2: We appreciate the reviewer’s suggestion, and relevant content has been supplemented.
Corresponding changes:
Page 2, line 54, "Rombouts et al found that Inconel 625 deposited using directed energy deposition also showed…" was corrected as "Rombouts et al found that Inconel 625 deposited using directed energy deposition (DED) also showed …".
Point 3: A deeper literature review on the application of AM in the repair of superalloys, in particular, In625 is necessary. Because for sure in the literature this aspect has been already considered and studied.
Response 3: We appreciate the reviewer’s suggestion on the literature review of the laser additive repairing Inconel 625 alloy.
Corresponding part that was added:
"The research work of LAR for nickel-based alloys mainly focuses on the repair defects, processing parameters, microstructures and mechanical properties of repaired samples. Onuike et al [26] used DED technology to repaired the internal cracks in Inconel 718 alloy. Sui et al [9] studied the tensile deformation behavior of LARed Inconel 718 alloy with a non-uniform microstructure."
Point 4: Following the last comment, the novelty aspect of this work should be highlighted.
Response 4: We appreciate the reviewer’s suggestion, and the relevant content has been supplemented.
Corresponding part that was added:
"To the best of our knowledge, there is no studies have investigated the microstructure and micro-mechanical properties of LARed Inconel 625 alloy around the repaired interface. "
Point 5: After the first introduction of the abbreviation of Laser Additive Manufacturing in the article, the full name should be replaced with the abbreviation.
Response 5: We appreciate the reviewer’s suggestion, and relevant content has been supplemented.
Corresponding changes:
Page 2, line 77, "The LAR experiments were performed on a laser additive manufacturing system, which…" was corrected as "The LAR experiments were performed on a LAM system, which…".
Page 3, line 82, "Then the defect was repaired layer by layer using the laser additive manufacturing system,…" was corrected as " Then the defect was repaired layer by layer using the LAM system,…".
Point 6: In the materials and methods, according to the template of the journal the brand of equipment together with the name of city and country should be inserted.
Response 6: We appreciate the reviewer’s suggestion, and relevant content has been supplemented.
Point 7: How many hardness points have been analyzed? What about the tensile tests?
Response 7: The Vickers hardness is tested at 3 points in different positions, while the nanoindentation is tested at 5 points in different zone, and tensile specimens are tested for 3 samples each.
Point 8: Page 12, line 279 there is an error for the references.
Response 8: We have corrected this error.
Point 9: If the tensile results are the average of some measurements, the results in Table 3 need standard deviation.
Response 9: We appreciate the reviewer’s suggestion, and relevant standard deviation has been supplemented.
Point 10: The conclusion section should be rewritten. First, the description of work should be inserted as a starting paragraph then the important findings of the work should be listed as conclusions.
Response 10: We appreciate the reviewer’s suggestion, and the conclusion section has been rewritten.
Corresponding part that was revised:
Conclusions:
"In this paper, Inconel 625 alloy substrates with premade trapezoidal groove shaped defects were repaired using LAR. The microstructures around the repaired interface and the corresponding mechanical properties were investigated. The conclusions are outlined below:
…"
